# Fast Blue and Cholera Toxin-B Survival Guide for Alpha-Motoneurons Labeling: Less Is Better in Young B6SJL Mice, but More Is Better in Aged C57Bl/J Mice

**DOI:** 10.3390/bioengineering10020141

**Published:** 2023-01-20

**Authors:** Hasan Farid, Weston B. Gelford, Lori L. Goss, Teresa L. Garrett, Sherif M. Elbasiouny

**Affiliations:** 1Department of Neuroscience, Cell Biology, and Physiology, Boonshoft School of Medicine, College of Science and Mathematics, Wright State University, Dayton, OH 45435, USA; 2Department of Biomedical, Industrial, and Human Factors Engineering, College of Engineering and Computer Science, Wright State University, Dayton, OH 45435, USA

**Keywords:** fast blue, cholera toxin subunit B, motoneuron, immuno labeling

## Abstract

Fast Blue (FB) and Cholera Toxin-B (CTB) are two retrograde tracers extensively used to label alpha-motoneurons (α-MNs). The overall goals of the present study were to (1) assess the effectiveness of different FB and CTB protocols in labeling α-MNs, (2) compare the labeling quality of these tracers at standard concentrations reported in the literature (FB 2% and CTB 0.1%) versus lower concentrations to overcome tracer leakage, and (3) determine an optimal protocol for labeling α-MNs in young B6SJL and aged C57Bl/J mice (when axonal transport is disrupted by aging). Hindlimb muscles of young B6SJL and aged C57Bl/J mice were intramuscularly injected with different FB or CTB concentrations and then euthanized at either 3 or 5 days after injection. Measurements were performed to assess labeling quality via seven different parameters. Our results show that tracer protocols of lower concentration and shorter labeling durations were generally better in labeling young α-MNs, whereas tracer protocols of higher tracer concentration and longer labeling durations were generally better in labeling aged α-MNs. A 0.2%, 3-day FB protocol provided optimal labeling of young α-MNs without tracer leakage, whereas a 2%, 5-day FB protocol or 0.1% CTB protocol provided optimal labeling of aged α-MNs. These results inform future studies on the selection of optimal FB and CTB protocols for α-MNs labeling in normal, aging, and neurodegenerative disease conditions.

## 1. Introduction

Retrograde tracing is a technique that exploits retrograde axonal transport, which uptakes a tracer along the axons connecting the infusion site (e.g., the muscle) to the distal source cell (e.g., the motoneuron in the spinal cord). Neuroanatomical tracers have been frequently used to label neuronal structures since their discovery in 1971 by Kristensson and Olsson [1]. Since then, numerous tracers of differing compositions have been developed, including dextran conjugates such as Fluoro-Ruby (tetramethyl rhodamine-dextran amine conjugate) [2], chemical tracers such as Fluoro-Gold [3], and enzymatic proteins such as horseradish peroxidase [1]. The usefulness of retrograde tracers is a) their ability to trace neural connections from synapses (i.e., their terminals) back to cell bodies (i.e., their sources), and b) their adaptability to different studies by changing the method of application and detection [4]. Additionally, tracers allow the identification of neuroanatomical pathways [5], improve our understanding of axonal transport mechanisms during states of health and disease [6], and facilitate the development of novel treatments for nerve injury [7]. As axonal transport becomes deficient with aging [8,9], assessing how well retrograde tracers label alpha-motoneurons (α-MNs) in aging is critical.

Of the various neuroanatomical tracers described in the literature, fast-blue (FB) and cholera toxin B (CTB) are retrograde tracers that have been extensively used in labeling MNs [10,11,12,13,14]. FB is a chemical fluorescent dye that emits blue light upon excitation [15,16]. CTB, on the other hand, is the beta-subunit of a bacterial toxin that is secreted by the bacterium *Vibrio cholerae* [17]. Although these tracers have several features in common, such as their retrograde transport ability and their fluorescence [13,14], they differ in some aspects. First, their uptake mechanisms are different in that CTB uses receptor-mediated endocytosis by binding onto monosialotetrahexosylgangliosides (GM1) located on the neuronal membrane. This increases the binding affinity of the tracer, leading to higher CTB uptake efficiency [16,17] though it is encased in vesicles, has a more granular appearance, and does not display morphology well [16]. FB, in contrast, is passively taken up by neurons and labels them through active transport using endosomes [16]. These different mechanisms of transport and uptake impact the number of labeled cells and the type of tracer best for different applications such as neurodegenerative or aging studies [18]. Second, their compositions are different: CTB is a bacterial toxin that can be conjugated, making it adaptable to any type of microscopy [19,20], whereas FB cannot be conjugated and is thus only useful for fluorescence microscopy [15,16]. FB’s fluorescence property is somewhat limiting because its blue fluorescence requires an Ultraviolet (UV) wavelength (360 nm) for excitation. This is a potential issue in cell culture studies because UV light can cause phototoxicity in labeled cells [16]. Additionally, FB can also interfere with cell adhesion in cell culture studies [16] and is limited in that it can uptake well only if the terminals are intact, which could affect outcomes in neurogenerative diseases [16].

One aspect common to FB and CTB tracers is that they have been extensively used in the literature in various protocols and wide ranges of concentrations (e.g., FB have been used from 0.5% to 5%, a 10-fold range). While high tracer concentrations would be expected to provide high labeling quality (i.e., brighter, more, and larger proportions of labeled cells), they also run the risk of tracer leakage from MNs to neighboring cells, thereby losing MNs specificity. Despite this tradeoff, no study has yet performed a systematic assessment of how various protocols and tracer concentrations of FB and CTB impact their neuronal labeling quality to optimize MN labeling. Without this knowledge, sub-optimal protocols and tracer concentrations would continue to be inadvertently used. Therefore, the overall goal of the present study is to assess the effectiveness of different protocols for the retrograde tracers FB and CTB to identify optimal protocols for labeling α-MNs in young and aged mice. Specifically, we compared the labeling quality of these tracers at standard (i.e., most commonly used) concentrations reported in the literature (FB 2% and CTB 0.1%) versus lower concentrations. To achieve that, three different FB concentrations (0.1%, 0.2%, and 2%—weight/volume) and two CTB concentrations (0.05% and 0.1%—weight/volume) were injected into hindlimb muscles in young wild-type (WT) B6SJL mice. We examined the labeling quality at two time-points, 3- and 5-days post-administration (i.e., the labeling duration between intramuscular injection and the terminal immunohistochemistry experiment). We compared seven parameters: Labeling intensity, the density of labeled cells, the volume of neurite projections, total length and longest path distance of labeled neurites, labeling specificity to α-ΜΝs, and tracer leakage. Our results showed that a low concentration, short labeling duration FB protocol provided optimal labeling of young α-MNs, whereas a high concentration, long labeling duration FB or CTB protocol provided optimal labeling of aged α-MNs. This study aimed to provide systematic, detailed assessments of multiple protocols for these two tracers. The results were then useful in the selection of optimal FB and CTB protocols for retrograde α-MN labeling in young and aged mice in aging and neurodegenerative disease studies.

## 2. Materials and Methods

### 2.1. Animals

For experiments in young mice, wild-type (WT) B6SJL mice breeders were purchased from the Jackson Laboratory (stock #002726) and a line was established at Wright State University (WSU). A total of 34 adult male young mice (6–7 weeks of age) obtained from the colony were recruited in the study and randomly assigned to ten experimental groups (N.B., however, 3 mice were excluded from the study, see Section 2.7 for detail). The B6SJL strain was chosen because transgenic mice of several neurodegenerative diseases (e.g., ALS and AD) were developed from this background [21,22,23,24]. Each group tests a given tracer at a different concentration and labeling duration and Table 1 shows the number of animals that contributed successful data to each group. Mice were housed under appropriate conditions at the WSU Laboratory Animal Resource (LAR) facility prior to surgeries with approximately 4 mice housed per cage with cotton bedding material in a 12-h light/dark cycle with water and food provided ad libitum. After surgeries, mice were then individually housed in single cages with a 12-h light/dark cycle with water and food provided ad libitum until euthanized. All experiments and procedures were conducted in accordance with the Guiding Principles for Research Involving Animals and Human Beings, and in compliance with federal and WSU guidelines whose Laboratory Animal Care and Use Committee (LACUC) has approved these experiments (approved protocol numbers: AUP 1045, 1196 and 1117).

For experiments in aged mice, 12 C57Bl/J (C57) mice (25 months) were obtained from the National Institute on Aging (NIA) and recruited in the study (see Table 1). This strain was chosen because it is the most common strain used in aging research [25,26,27,28]. Housing and cages were the same as described above. To test the two tracers (FB and CTB) at different concentrations and labeling durations in aged animals, mice were distributed among several groups as shown in Table 1. Each aged animal has undergone one surgery during which it was injected with FB in one limb and CTB in the other limb to measure the difference between tracers. This was performed to minimize the number of animals used in these experiments and to robustly evaluate each tracer in each animal, as each animal contributes equally to the FB group and CTB groups. Because 0.1% CTB and 2% FB are the standard concentrations in the field [29,30], they are the reference (or control) groups in our young and aged group comparisons.

### 2.2. Surgical Procedures and Tracer Injections

Surgical procedures were conducted in the morning to mid-afternoon in WSU’s LAR sterile surgical suite, and each animal underwent one surgery. Mice were anesthetized with isoflurane at 3–5% for induction, then maintained at (2–3%) during surgery via nose cones. Four hindlimb muscles—soleus (Sol), tibialis anterior (TA), and lateral and medial gastrocnemius (LG and MG)—were exposed by a small incision and separation of the overlying biceps femoris muscle. In a given mouse, all four muscles were injected with one of the following tracer/concentration protocols: (1) 5 µL of Fast-Blue (FB) (Polyscience, Warrington, PA, USA catalog 17740-1) at (i) 0.1%, (ii) 0.2%, or (ii) 2%; in weight/volume; or (2) 5 µL of CTB-488 Alexa Fluor conjugate (Invitrogen, Waltham, MA, USA catalog C22841) at (i) 0.05% or (ii) 0.1%; in weight/volume. Injections were given through a 10 μL Hamilton syringe with a 33-gauge needle. Overall, each mouse received a total of 20 µL of one type/concentration of tracer injected into its four hindlimb muscles. Intraoperative monitoring was conducted every 5 min based on movement, respiration, and color. Buprenorphine (0.0025 mL/g) was injected subcutaneously immediately after surgery followed by subcutaneous injection of Carprofen (0.01 mL/g) 24 h after surgery for post-operative pain relief. Mice were then euthanized and perfused 3- or 5-days post-injection of tracers (labeling duration).

### 2.3. Perfusion and Dissection of Spinal Cord

All perfusions and dissections were conducted at the WSU Microscopy Core Facility perfusion room. All mice were anesthetized with a lethal dosage of Euthasol solution (150 mg/kg, pentobarbital sodium, and phenytoin sodium) via intraperitoneal injection, either 3 or 5 days after injection of retrograde tracers into hindlimb muscles. After confirming the lack of reflexive response via toe pinch, mice were transcardially perfused with vascular rinse (0.01 M phosphate buffer with 0.5% NaCl, 0.025% KCl, and 0.05% NaHCO_3_, pH 7–8), followed by 4% paraformaldehyde in 0.1 M phosphate buffer, pH 7–8. After fixation, mice had their spinal cord extracted from the mid-thoracic to the early sacral region. These extracted spinal cords were submerged into 4% paraformaldehyde for ~2 h before being transferred into 15% (weight/volume) sucrose solution at 4 °C overnight.

### 2.4. Identification of Spinal Cord Segments and Sectioning

Extracted spinal cords were removed from 15% sucrose and pinned onto a Slyguard^®^ (Electron Microscopy Sciences, Hatfield, PA, USA) padded dissection petri dish with large insect pins. Smaller insect pins were then used to mark the origins of the ventral roots from L3 to S1 (lower lumbar spinal cord region). After identifying the lower lumbar spinal cord region, ventral roots were cut, and spinal cord segments were stained using marking dyes (Bradley Products, Bloomington, MN, USA) with contrasting colors to identify each of the individual lumbar spinal cord segments during sectioning. Two transverse cuts were made at L2 and at S2. The lower spinal cord regions were then placed into rubber molds with Tissue Freezing Medium™ (General Data, Cincinnati, OH, USA, catalog TFM-C) and frozen with cold isopentane. Frozen tissue blocks were removed and stored at −80 °C until sectioning. Frozen tissue blocks were transversely sectioned at 45 µm at ~−25 °C on an HM 550 ThermoFisher^®^ Cryostat (Waltham, MA, USA). Tissues were serially collected from L3 to L6 in 24-well plates filled with cryoprotectant.

### 2.5. Mounting and Immunohistochemistry

Approximately 3 days after perfusion, ~5–6 sections were collected from each spinal cord segment and transferred into Netwell^®^ (Electron Microscopy Sciences, Hatfield, PA, USA) inserted 12 well-plates. Transverse sections were washed with 1× Phosphate Buffered Saline solution (PBS), pH 7.4 (ThermoFisher^®^ Scientific Inc., NJ, USA catalog 10010023) 3 times at 10-min intervals. This was followed by washing once in cupric sulfate (10 mM Cupric sulfate in 50 mM ammonium acetate) solution for 45 min to prevent the autofluorescence of endogenous protein, lipofuscin, within neurons. Sections were then rinsed in DDI-filled NetWell^®^ 12 well-plates (Electron Microscopy Sciences, Hatfield, PA, USA), followed by another minute of PBS washing before being mounted onto positively charged microscope slides and cover-slipped with Vectashield^®^ antifade mounting medium (Vector Laboratories, Newark, CA, USA, catalog H-1000). This process was repeated for all sections. Additional sections from 2% FB 3-day and 0.1% CTB 3-day were labeled with Choline Acetyltransferase (ChAT) and Vesicular acetylcholine transporter (VAChT) to determine if the tracers were labeling cholinergic MNs. For this staining, sections were washed 3 times with PBS-T (0.01 M PBS containing 0.1% Tritron-X, pH 7.3) followed by blockage with normal horse serum (10% PBS-T) for an hour. Sections were then incubated with the primary antibody, ChAT (mouse antibody, Novus Biologicals, catalog #NBP2-46620, Centennial, CO, USA, RRID: AB_2922998) at 1:100 dilution in PBS-T overnight at 4 °C. Additional sections were also labeled with VAChT (mouse antibody, Novus Biologicals, catalog #NBP2-59378, Centennial, CO, USA, RRID: AB_2922997) at a 1:400 dilution in PBS-T and incubated at 4 °C overnight. The following day, the Alexa Fluor^®^ 647 anti-mouse secondary antibody (Jackson Immuno Research Inc., catalog #715-605-150, West Grove, PA, USA, RRID: AB_2340862) was diluted to 1:100 with PBS-T and sections were incubated for ~2 h before being mounted onto positively charged microscope slides and coverslipped in Vectashield^®^ antifade mounting medium.

As small, round, and blue fluorescent dots were seen in FB protocols (0.1% 5-day, 0.2% 5-day, 2% 3-day and 5-day), additional staining with VAChT and NeuN labels was conducted in these protocols to determine if these small and round structures are neuronal or non-neuronal (i.e., NeuN determines if these dots are neuronal, whereas VAChT determines if they are MNs). On the first day of staining, tissue sections were washed 3 times with PBS-T (0.01 M PBS containing 0.1% Tritron-X, pH 7.3) followed by blockage with normal horse serum (10% PBS-T) for an hour. Sections were then incubated with primary antibodies, VAChT (mouse antibody, Novus Biologicals, catalog #NBP2-59378, Centennial, CO, USA, RRID: AB_2922997), at a 1:400 dilution in PBS-T and incubated overnight. The following day, the Alexa Fluor^®^ 647 anti-mouse secondary antibody (Jackson Immuno Research Inc., catalog #715-605-150, West Grove, PA, USA, RRID:AB_2340862) was diluted to 1:100 with PBS-T for approximately 2 h. After that, the sections were washed 3 times with PBS-T once more before the primary antibody, NeuN, (guinea pig antibody, Millipore, catalog# ABN90, St. Louis, MO, USA, RRID: AB_11205592), at 1:300 in PBS-T was applied and left to incubate overnight. The next day, Alexa Fluor^®^ 488 anti-guinea pig secondary antibody (Jackson Immuno Research Inc., catalog #706-545-148, West Grove, PA, USA, RRID: AB_2340472) was diluted to 1:100 with PBS-T for approximately 2 h.

In aged C57 tissue, sections were labeled with VAChT (goat antibody, Millipore, catalog #ABN100, St. Louis, MO, USA, RRID: AB_2630394) and the Alexa Fluor^®^ 647 anti-goat secondary antibody (Jackson ImmunoResearch Inc., catalog #705-605-147, West Grove, PA, USA, RIDD: AB_2340437) using a similar protocol as above. In addition, prior to mounting, the sections were washed in a cupric sulfate buffer to quench the autofluorescence due to lipofuscin accumulation. Afterwards, sections were mounted onto positively charged microscope slides and coverslipped in Vectashield^®^ antifade mounting.

### 2.6. Imaging and Data Analysis

Confocal imaging was performed 4 days post-fixation in all protocol groups using an FV1000 Olympus confocal microscope objective lens at 20× with 1-µm z-steps. Only complete sections that displayed both ventral horns without any major tears were imaged. The primary motor pools (laminae IX) in the ventral horn of each slice were imaged for analysis. The images were taken at 1024 × 1024 in resolution and 1.2× in zoom (528 × 528 µm). Fluoview image analysis software (Olympus Corporation, Pittsburgh, PA, USA) was used to measure the labeling intensity ratio, labeling intensity difference, density of labeled cells, and percentage of non-neuronal co-labeling from images.

To quantify the tracer staining intensity relative to the background, two intensity calculations were performed on each image: (1) Labeling intensity ratio and (2) labeling intensity difference. With the labeling intensity ratio, the average labeling intensity of a labeled α-MN is divided by the average background intensity. With the labeling intensity difference, the average background intensity is subtracted from the average labeling intensity of a labeled α-MN. The background intensity was measured from an area on the slice that did not have labels on it. For FB protocols, intensity values were obtained by circling the largest cross-sectional area (LCA) of a soma, whereas CTB protocol intensity values were obtained by circling bright vesicles that appeared in the soma of a CTB labeled α-MN. The reason for this discrepancy in measuring the intensity value is that CTB utilizes receptor-mediated endocytosis in its uptake mechanism, and therefore the CTB tracer is seen encapsulated in a vesicle within the soma, whereas FB labels the cytoplasm of a soma. This analysis allows FB and CTB tracer protocols to be compared equally without the influence of their uptake mechanism. Therefore, using the two different intensity calculations allows a thorough investigation of the measure, and the different measurements account for the difference in background fluorescence.

The density of labeled cells was calculated by counting the number of labeled MNs within the 3D z-stack of 20x images and dividing their number by the total volume of the 3D z-stack images, thereby allowing comparison of how many MNs are labeled among tracer protocols.

As non-MN labeling was seen as small, round, and blue fluorescent dots in some FB protocols, NeuN was used to determine if these dots are neuronal and VAChT to determine if they are MNs. The co-labeling percentage was obtained by counting the number of FB-labeled dots co-labeled by NeuN only and dividing it by the entire count of FB-labeled dots in selective FB protocols.

Neurolucida^®^ 360 (MBF Bioscience, Williston, VT, USA) image analysis software was used to measure the 3D properties of labeled somas and neurites in the young mice. A neurite was defined as any projection out of the soma of an α-MN, as we could not determine if these projections were dendrites or axons without additional labeling. Analysis of neurites commenced with identifying somas, then using Neurolucida^®^ 360 software to label neurites connected to their respective somas. Neurolucida^®^ 360 Explorer software (MBF Bioscience, Williston, VT, USA) was then used to obtain three measurements: (1) Neurite volume (µm^3^), (2) total neurite length (µm), and (3) the longest neurite path distance (µm). These parameters were selected because they assess different aspects of neuronal labeling quality by tracers. For instance, neurite volume provides a measure of how well a tracer fills the 3D structure of neurites, which is useful in studies aiming to reconstruct anatomical morphologies. Total neurite length was calculated as the summation of lengths of labeled neurites branching out of somas, which provides a measure of how well a tracer labels somatic primary projection. The longest neurite path distance was calculated as the longest path formed by labeled neurites away from the soma, which provides a measure of how far away from the soma a tracer is capable of labeling neurites. Previous literature has shown that MNs with an LCA area equal to or greater than 300 μm are deemed to be α-MNs [31,32]. Therefore, in this study, those labeled MNs with an LCA area less than 300 μm are not included in these measurements, as they are not deemed to be α-MNs.

### 2.7. Statistical Analysis and Data Presentation

SPSS^®^ (IBM Corporation, Armonk, NY, USA) statistical software was used for the statistical analysis of all data. Prism GraphPad (GraphPad Software, Boston, MA, USA) was used for all graphing needs for this study. Data for all measurements were found not normally distributed and have unequal variance as indicated by Levene’s test and, therefore, non-parametric statistical analysis was conducted. Each experimental group (tracer, concentration, and labeling duration) was coded and tested with the Kruskal–Wallis test and Dunn’s post-hoc test. The threshold for significance (α) for all statistical analyses was 0.05. Any data with a *p*-value > 0.05 was deemed not statistically significant (N.S). All data in the figures are shown as median ±95% confidence intervals.

Three mice were excluded from the statistical analysis due to insufficient tracer labeling. These mice were part of the 0.2% FB 3-day and 0.05% CTB 3-day groups and had less than 10 labeled cells in total. This is significantly less than other animals in the groups and what is expected if the tracer has been successfully taken up at the muscle and retrogradely transported back to the spinal cord. Therefore, it was concluded that the tracer labeling was faulty in some way in these mice and their data were excluded. The sample sizes listed in Table 1 show the number of animals that contributed successful data to each group and do not include the excluded animals. For each group, the data from each animal were compared, and we confirmed that animals contributed comparably to the collected total sample.

## 3. Results

### 3.1. Lower Concentration/Short Labeling Duration FB and CTB Protocols Are as Effective as Higher Concentration/Long Labeling Duration Protocols in Young (6–7 Weeks) Mice

The goals of the present study are:(1) to assess the effectiveness of different protocols of the retrograde tracers FB and CTB in labeling spinal α-MNs, and (2) to compare the labeling quality of tracers’ standard concentrations (FB 2% and CTB 0.1%) versus lower concentrations, in an effort to avoid common issues such as leakage. To achieve that, several measurements were compared among experimental groups representing different tracers, concentrations, and labeling durations (see Table 1 for a summary of the experimental groups). First, we compared the labeling intensity ratio among the experimental groups as shown in Figure 1A. The data showed that 0.2% FB 3-day provided the highest labeling intensity for FB and 0.1% CTB 3-day provided the highest labeling intensity for CTB (Figure 1A). When the intensity ratios of 3-day and 5-day FB and CTB protocols were compared with each other, neither tracer protocol was significantly different (Figure 1A). Additionally, 5-day FB and CTB protocols had generally similar or lower labeling intensity than 3-day protocols (*p* < 0.001), except for the 0.1% FB protocol, which showed the opposite trend (Figure 1A). Importantly, the labeling intensity of 0.2% FB was not statistically different from that of the standard 2% FB concentration, in either 3-day or 5-day protocols. This indicates that a 10-fold reduction of the standard FB concentration is equally effective in labeling α-ΜΝs. Similarly, the lower concentration of CTB was also not statistically different from that of the standard 0.1% CTB at 3-day and 5-day protocols, confirming that a lower CTB concentration is equally effective in labeling spinal MNs.

Because the average background intensity could influence the outcome of the labeling intensity ratio, a labeling intensity difference measure was also analyzed among all tracer protocols (Figure 1B). The results of this measurement show that FB protocols have greater labeling intensity than CTB protocols (*p* < 0.01), except for 0.1% CTB 5-day (Figure 1B). Similarly, FB and CTB protocols of higher tracer concentrations or a longer labeling duration did not show higher labeling intensity (Figure 1B). In sum, our results show that in young mice, (1) FB and CTB provide comparable labeling intensity of spinal MNs, (2) protocols of longer labeling duration (5-day) mostly give similar, sometimes lower, labeling intensity than shorter (3-day) protocols, and (3) lower-concentration FB and CTB protocols provide equal, sometimes higher, MN labeling intensity than that of the higher standard concentrations.

### 3.2. CTB Is More Effective in Labeling More α-MNs of Young Mice

To assess how successfully FB and CTB tracers are retrogradely transported from muscle fibers to the spinal cord, we compared the number of α-MNs labeled among the experimental protocols. We injected FB and CTB tracers into multiple hindlimb muscles (Sol, TA, MG, and LG)—to maximize the number of labeled cells and to achieve equal distribution of labeling across the lumbar spinal cord region [33]—and measured the density of labeled α-ΜΝs (number of labeled α-ΜΝs in 3D z-stacks per unit tissue volume). Our data showed statistically significant differences in the density of labeled α-ΜΝs among different FB and CTB protocol concentrations (*p* < 0.01). Specifically, the 0.05% CTB 3-day protocol had significantly higher labeled cell density than many FB protocols (*p* < 0.001 and *p* < 0.05), indicating that CTB generally labels more MNs than FB. Furthermore, 3-day protocols generally had similar or higher cell density than 5-day protocols for both tracers (Figure 2, compare red to blue bars in all groups). With respect to tracer concentrations, FB protocols of low concentrations (0.1% and 0.2%) had comparable labeled cell density to that of the higher 2% FB standard concentration, and similarly, the lower 0.05% CTB 3-day protocol had a comparable labeled cell density to that of the 3-day higher 0.1% CTB standard concentration (Figure 2). These data further show no advantage for tracer protocols of higher concentrations or longer labeling durations than those of lower concentrations and shorter labeling durations. Together, these data show that (1) CTB generally labels more spinal MNs than FB, (2) 3-day protocols are as effective as 5-day protocols, and (3) protocols of low concentrations are as effective as, or sometimes better than, high-concentration protocols.

### 3.3. FB and CTB Label α-ΜΝ Anatomy of Young Mice Comparably

To assess how well FB and CTB label the anatomy of α-ΜΝs, we quantified and compared the 3D morphological properties of labeled α-ΜΝs, including their somas and neuronal projections (neurites) among the experimental groups. To achieve that, we used Neurolucida^®^ 360 software to measure three parameters: (1) Neurite volume (Figure 3), (2) total neurite length (i.e., the total sum of neurite length, which would be = L1 + L2 + … + L10 in Figure 4A,B), and (3) longest neurite path distance (which would be = L4 in Figure 4A,B). These parameters were selected because they assess different aspects of the 3D neuronal labeling quality. For instance, neurite volume provides a measure of how well the tracer fills the 3D structure of neurites. The total neurite length provides a measure of how many neurites are labeled by the tracer and how well the tracer labels neurites along their path. The longest neurite path distance provides a measure of how far a tracer is capable of labeling neurites away from the soma. Because a cell located near the edge of a section could have some of its neurites transected, thereby underestimating its neurites measurements, we, therefore, excluded cells located close to the section edge from the total and longest neurite length analysis.

For neurite volume, our data showed that regardless of concentration, 5-day FB protocols had higher labeling of neurites volume than 3-day FB protocols and higher than all CTB protocols (Figure 3). Specifically, a statistical difference was seen between 3-day and 5-day protocols for all FB protocols (*p* < 0.05) but not for CTB tracers. With respect to tracer concentrations, 0.1% and 0.2% FB protocols were not statistically different from the higher 2% standard FB concentration (3-day or 5-day protocols), and similarly, the 0.05% CTB protocols were not statistically different from the higher 0.1% standard CTB concentration (Figure 3, the last four bars).

With the total neurite length analysis, we continued to see similar trends with 3-day and 5-day protocols for both tracers having comparable total labeled neurite length (no statistical significance was noted between any blue and red bars at a given concertation in Figure 4C) and protocols of low and high concentrations with comparable total labeled neurite length (no statistical significance was noted among blue or red FB bars, or among blue or red CTB bars in Figure 4C). Between FB and CTB, protocols of both tracers had comparable total labeled neurite lengths, but 0.05% CTB protocols tended to show the lowest total neurite length values, whereas 0.1% FB 5-day and 0.1% CTB 5-day tended to show the highest total neurite length values (Figure 4C). When the longest neurite path length was compared among the experimental groups, no statistical difference was seen across all FB or CTB protocols. Importantly, the trends observed within total neurite length and longest neurite path distance were similar when cells with neurites close to the edge of the section were added to the analysis. Taken collectively, these results indicate that (1) 5-day FB protocols are the most effective in labeling the neurite volume, (2) low-concentration FB and CTB protocols are as effective as high-concentration protocols in labeling neurite volume, total neurite length, and longest neurite path distance of α-MNs, and (3) short labeling duration FB and CTB (i.e., 3-day) protocols are as effective as long labeling duration (i.e., 5-day) protocols in labeling neurite volume, total neurite length, and longest neurite path distance of α-MNs.

### 3.4. FB and CTB Label MNs, but Not Ins

As retrograde tracers, FB and CTB are expected to label α-MNs, but they could also label γ-MNs or interneurons (Ins) if transported via synapses. C-boutons are more likely to be found only on α-MNs but can also be on γ-MNs but not on Ins [34,35,36]. To determine if the labeled neurons are MNs and that only α-MNs were analyzed in this experiment, we stained spinal tissue with the ChAT antibody to label C-boutons and measured the LCA of all labeled MNs, removing those less than 300 μm (see Methods for details). We focused on 2% FB 3-day and 0.1% CTB 3-day protocols for ChAT labeling, as the high labeling intensity of these protocols maximizes the accuracy of this analysis (see Figure 1), thereby enhancing the rigor of this investigation. Our confocal images and analysis showed that 100% of neurons labeled with FB or CTB also showed ChAT co-labeling (Figure 5). The results of ChAT labeling support that FB and CTB label MNs only, when intramuscularly injected. Importantly, similar results were obtained when VAChT—another specific C-bouton antibody to label ΜΝs—was used in a separate tissue, confirming that FB and CTB label ΜΝs only.

### 3.5. Tracer Leakage with some FB Protocols

Because FB has been shown to leak from labeled MNs to other cells in the spinal cord [16], we examined images of all experimental groups for potential leakage effects. Our analysis showed the consistent appearance of FB leakage as small, round, and blue fluorescent dots that appear to be non-MN in the standard 2% FB concentration (both 3-day and 5-day protocols, see the white arrows in Figure 6A) and in the lower concentration of FB at 5-day. To determine if these small, round, and blue fluorescent dots could be neuronal and/or MNs, we stained some FB sections from 0.1% FB 5-day, 0.2% FB 5-day, and 2% FB 3-day and 5-day with NeuN and VAChT. After staining, a NeuN co-labeled percentage was obtained from these sections without the inclusion of the dots that were also co-labeled with VAChT. This percentage depicts whether or not these dots are neuronal cells. From our results in Table 2, it is seen that the majority of FB protocols, except for 0.2% FB 5-day, had a small percentage of NeuN co-labeling suggesting that these small, round, blue fluorescent dots are indeed not MNs and are not neuronal. Furthermore, we also observed the presence of a halo-like- effect specifically within 2% FB images, which appeared along with non-neuronal cell labeling. Therefore, our NeuN analysis confirms that the small, round, and blue fluorescent dots are non-neuronal and likely due to leakage from FB labeled MNs in the 2% FB protocols and, in some cases, a lower concentration of FB at 5 days only (Table 2). Interestingly, there was no non-MN cell labeling or the presence of halo-like effects with any concentration of CTB at 3 days or 5 days or with lower concentrations (<2%) of FB at 3 days. Collectively, the results of these experiments and the experiments in the previous section on FB and CTB specificity in labeling α-MNs show that (1) CTB and lower FB concentration (i.e., <2%) protocols at 3 days label α-MNs only without tracer leakage and (2) FB at higher concentrations (long or short labeling duration) or lower concentration/longer labeling duration protocols can exhibit tracer leakage leading to the labeling of additional non-neuronal labeling, in addition to the appearance of halo-like effects.

### 3.6. Intensity and Density Are Altered by Concentration of FB and CTB Labeling in Aged C57 Mice

While the first part of this study was performed in young mice, the goal of this part is to assess how well FB and CTB are retrogradely transported from muscle fibers to the spinal cord in aged mice, given that alterations in axonal transport evolve with aging [9,37]. Because data from young mice showed comparable labeling among tracer protocols of different concentrations and labeling durations, we, therefore, tested two concentrations (one low and one high) per tracer in aged mice.

For labeling intensity, both the intensity ratio and intensity difference relative to background were examined. Opposite to the trend in young mice, higher tracer concentrations appeared to work better than lower concentrations based on both intensity measures (Figure 7). This was true for FB protocols in Figure 7A and FB and CTB protocols in Figure 7B. Based on the intensity ratio measure, 2% FB and all CTB protocols appear to label α-MNs well and comparably (Figure 7A), whereas FB 2% appears to be the only protocol that labeled α-MNs best based on the intensity difference measure (Figure 7B). Therefore, FB at 2% appears to be a good protocol that labels α-MNs well regardless of how labeling intensity is measured (Figure 7). While labeling duration did not appear to impact the labeling intensity in most protocols, 2% FB always labeled best with the longer 5-day protocols regardless of how labeling intensity is measured (Figure 7). In sum, in aged mice, (1) higher tracer concentrations yielded higher labeling intensity, and (2) FB at 2% with a 5-day labeling duration appears to be a good protocol for labeling α-MNs regardless of how labeling intensity is measured.

The density of labeled α-MNs between the different experimental protocols was compared. The data showed a similar trend to the intensity measures in that higher tracer concentrations worked better and labeled more α-MNs in aged mice (*p* < 0.001, Figure 8). This was true for FB and CTB protocols as the 2% FB showed a higher density than 0.1% FB and the 0.1% CTB showed a higher density than 0.05% CTB (Figure 8). Labeling duration appeared to have no impact on the density of labeled α-MNs (Figure 8). Together, higher tracer concentrations yielded a higher density of labeled α-MNs in aged animals.

### 3.7. FB and CTB Labels αMN’s Differently in Aged C57 Animals

To evaluate how well FB and CTB labeled the anatomy of α-MNs, we quantified the neurite volume, total length, and longest path distance in aged animals. As lower tracer concentrations had lower labeling intensity, which could affect the accuracy of the morphological measurements, we only tested the highest concentration of both tracers: 2% FB and 0.1% CTB. For the neurite volume, opposite to young animals, CTB worked better than FB (*p* = 0.02) and the labeling duration did not seem to influence the labeling (Figure 9A). For the neurite total length and longest path distance, both tracers had comparable labeling (i.e., no statistical significance was noted) with no apparent influence of labeling duration (Figure 9B,C). As lipofuscin accumulates in the tissue with aging, and despite our efforts to quench the autofluorescence resulting from lipofuscin accumulation (see Methods), the remaining autofluorescence did not allow us to accurately determine what was leakage and what was background staining. Thus, tracer leakage was not assessed in aged mice. Collectively, CTB appeared to be better at labeling the morphology of α-MNs in aged animals than FB with no effect on labeling duration.

## 4. Discussion

This study provides, for the first time, a systematic assessment and comparison of FB and CTB under different experimental conditions, such as tracer concentrations, labeling duration, and age. These two retrograde tracers are widely used in labeling MNs and are important in understanding MN labeling in neurodegenerative diseases, most of which occur in old age. Seven different aspects of neuronal labeling quality were examined: (1) Labeling intensity ratio and difference, (2) density of labeled cells, (3) volume of labeled neurites, (4) total length of labeled neurites, (5) longest path distance of labeled neurites, (6) labeling specificity to ΜΝs, and (7) tracer leakage through NeuN co-labeling analysis. A summary of how each tracer protocol performed across all parameters in young and aged mice is provided in Table 3. Generally, less (tracer concentration and labeling duration) was better in labeling young α-MNs, but more was better in labeling aged α-MNs. FB appeared to be a good tracer for labeling young (6–7 weeks) or aged (25 months) α-MNs, but CTB was a good alternative for labeling aged α-MNs. Accordingly, these results provide a useful guide to selecting optimal protocols when using FB or CTB retrograde tracers to label α-MNs in normal, aging, and neurodegenerative disease conditions.

### 4.1. Summary of Findings

The present study examined two strains of mice: The B6SJL strain for young mice and the C57Bl/J strain for old mice. The B6SJL strain was chosen for the young experiments because transgenic mice of several neurodegenerative diseases, such as ALS and AD, are young and come from this background strain [21,22,23,24], whereas the C57Bl/J strain was chosen for the aged experiments because it is the most common strain used in aging research [25,26,27,28]. Thus, our results are relevant to tracer studies in both ALS and aging research. Additionally, because sex is a biological variable in ALS, AD, and aging, this study focused only on male mice and our future work will study female mice. Given the difference in strains, our young and aged results were not directly compared to each other but compared within each age group.

Table 3 shows a summary of our findings on tracer protocols for labeling α-MNs in young and aged mice. In young mice, low FB and CTB concentrations provided comparable labeling to the 10-fold 2% FB and 2-fold 0.1% CTB standard concentrations. For most morphological measurements, except volume, the labeling duration did not impact the labeling quality (i.e., protocols with long labeling duration were as effective as those with short labeling duration). As a tracer, FB performed comparably or better than CTB for most labeling parameters, except cell density. However, FB protocols suffer tracer leakage, except at low concentrations and short labeling durations. Together, a FB protocol with low tracer concentration and short labeling duration would provide optimal labeling of α-MNs in young mice.

In aged mice, the qualities of an optimal tracer protocol reversed from those in young mice. Higher tracer concentrations labeled α-MNs better than lower concentrations. Protocols with a long labeling duration always labeled α-MNs well for all morphological measurements (short labeling duration protocols did not always perform well). FB performed comparably or better than CTB for most labeling parameters, except volume. However, as leakage was not assessed in aged mice due to accumulated lipofuscin, the presence of leakage is not excluded in high-concentration FB protocols in aged mice. Thus, CTB, which does not suffer leakage, could be a good alternative in labeling aged α-MNs given that CTB performed generally well, except for tracer intensity.

One inherent limitation of this study is that the sample size of our measurements varied among the different tracer groups. This limitation resulted from the variation among the tracer protocols in how well they labeled the tissue. Thus, labeling intensity was one important factor in determining how many accurate measurements we can collect from an image (i.e., the higher the labeling intensity, the larger accurate measurements we can collect from an image). Because of that, tracer concentrations with high labeling intensity (e.g., 2% FB protocols in young mice) usually had measurements of a larger sample size than those with low labeling intensity (e.g., 0.05% CTB protocols in young mice). Because the goal of this study is to compare the labeling quality of different tracer concentrations, the variation in sample sizes among the different protocols was a limitation that was hard to avoid. To mitigate the effects of this limitation, we relied on power analysis and our prior experience in collecting a number of accurate measurements per group exceeding the minimum needed sample size of that cell parameter (N.B., minimum sample sizes differ among the cell properties depending on the biological variability within each parameter). Importantly, comparisons that showed statistical significance in this study did not have drastic variation in the sample size between the groups, indicating that the primary conclusions of this study were not affected by the variability among sample sizes.

### 4.2. High or Low Tracer Concentration?

Although high tracer concentrations would be expected to provide high labeling quality, they also result in tracer leakage, thereby losing the labeling specificity of the desired neuronal populations. This tradeoff makes the selection of FB and CTB protocols and concentrations particularly challenging because these tracers have been used in literature in various protocols and a wide range of concentrations with no study highlighting the advantages and disadvantages of different tracer concentrations or suggesting optimal protocols. Because of the absence of this knowledge, sub-optimal protocols and high tracer concentrations continue to be used in various studies ranging from injecting tracers into muscle, nerves, and other locations of the rodents’ body to labeling α-MNs and other various types of neurons. For instance, in the last seven years only, at least 13 studies have used high FB concentrations (>1.5%, and as high as 5%) [38,39,40,41,42,43,44,45,46,47,48,49,50,51] while only 6 studies have used low FB concentrations (<1.5%) [52,53,54,55,56]. This indicates that low FB concentrations (i.e., <1%) are still not popular, and their advantages over high concentrations are still unknown despite having been used for a long time [57,58,59]. Data from the present study fill this knowledge gap and provide a direct comparison of the effects of high versus low tracer concentrations on the various aspects of neuronal staining quality and, therefore, guide the selection of optimal tracer protocols and concentrations. Moreover, the large differences in tracer protocols’ desired characteristics needed for optimal labeling of young versus aged MNs highlight the importance of the tracer choice, its concentration, and labeling duration. In other words, there is no one optimal tracer protocol that will label MN at all ages, and protocol selection will depend on the experimental design. Such information is critical for aging studies in which MNs of various ages (typically young and old) are labeled, and for neurodegenerative disease studies, such as amyotrophic lateral sclerosis (ALS), in which MN properties are tracked over the lifespan of the animal (i.e., at young and old ages) to study MN excitability dysregulation [60,61].

### 4.3. Optimal Timing, Is Shorter Better?

It has been suggested that FB is the tracer of choice as compared to Fluoro-Gold (FG) and dextran conjugate tracers (Mini-Ruby, Fluoro-Ruby, and Fluoro-Emerald) because FB labels a high number of MNs with persistent quality in labeling intensity for up to 24 weeks of labeling duration [62]. Our results confirm the positive characteristics of the FB tracer, but also establish that CTB is better than FB in staining younger α-MNs and in staining the volume of aged α-MNs (Table 3). Interestingly, our results for labeling intensity ratio contradict some studies in the literature that showed consistent FB labeling intensity with protocols of longer labeling duration from 8 weeks to 24 weeks [29,62]; however, the intensity was not quantified in these studies. The only exception to this was in 0.1% FB in young animals and 2% FB in old animals, in which the labeling intensity was higher for the 5-day protocol than the 3-day protocol. Additionally, for the young mouse measurements, 3-day protocols were generally comparable to, and sometimes better than, 5-day protocols. In aged mice, protocols of a longer labeling duration may be better at labeling α-MNs (Table 3).

### 4.4. Concentration Is Age Dependent

In intramuscular injection studies that utilize rodents such as ours, the standard concentration at which FB has been used is mostly 2% [48,63]. Interestingly, our results show that a 0.2% FB protocol—a much lower concentration than what has been used before and 10-fold lower than the standard concentration—has comparable labeling quality in young mice (see Figure 1), while in aged mice, 2% was clearly better (Figure 7 and Figure 8). This was also true for CTB as a two-fold reduction in the standard 0.1% concentration did not change the labeling quality in young mice (Figure 1), but higher concentrations showed improved labeling quality in aged mice (Figure 7 and Figure 9). In addition to the economic advantage of using less tracer, low FB concentrations avoid leakage onto other non-MN cells and non-neuronal cells, as well as the appearance of halo-like effects in images, two risks of higher tracer concentrations (Figure 5 and Figure 6). To our knowledge, only two studies have described the appearance of accidental neuronal staining caused by FB leakage in facial rat motoneurons [29,64]. In addition, the appearance of halo-like effects by FB is consistent with the literature regarding other fluorescent tracers [16]. The likely explanation for these effects is that they result from FB leaking from labeled neurons [16]. Interestingly, from our NeuN analysis, the majority of these non-MNs cells are not neuronal as there was a low percentage of NeuN co-labeling. In addition to our analysis, we also found that those that are co-labeled with NeuN could simply be dendrites from MNs whose somas are not located in the section. Thus, the finding that significantly lower FB concentrations at 3 days have comparable neuronal staining quality to that of higher concentrations while avoiding leakage, is novel to this study. On the other hand, lowering the CTB concentration had adverse effects on α-MNs’ labeling intensity relative to the standard CTB concentration (Figure 1). Although previous studies examined different labeling durations of the 0.1% CTB concentration [30], there are no data in the literature comparing the effects of different CTB concentrations on the quality of neuronal labeling. Thus, the results provided by the present study constitute an original contribution to the literature. In sum, lowering the FB concentration does not decrease neuronal labeling quality in young animals but does avert non-α-MN cell labeling and halo-like effects. Conversely, lowering the CTB concentration decreases neuronal labeling quality.

In our study, projections of labeled MNs were called neurites because we could not determine if these projections were dendrites or axons without additional labeling. Our results show that FB protocols are generally effective in labeling the neurite properties of α-MN in young mice, whereas both FB and CTB are good options in aged mice (Table 3). Notably, in young mice, CTB was found to label long neurites significantly better than Fluorogold [20], which has been speculated to be potentially harmful to labeled neurons in the long term [65]. This effectiveness of FB and CTB tracers in labeling somas and neurites renders them useful in studying neurodegenerative diseases, such as ALS, in which α-MNs experience changes in size [60] or sarcopenia, in which aged individuals show neuromuscular junction loss [26]. However, FB and CTB labeling are only good for measuring the morphological properties of the α-MN soma and primary projections: In our images, all neurite projections from labeled α-MNs had few or no branches. Thus, intracellular fillings—as opposed to intramuscular fillings via retrograde tracers—would be the method of choice to study the full dendritic anatomy of α-MNs.

## 5. Conclusions

This paper examined two retrograde tracers commonly used in research: FB and CTB were systematically assessed under different experimental conditions, such as varying age, concentrations, and labeling durations. Several observations emerged from our analysis: Protocols of lower tracer concentration and shorter labeling duration were generally better in labeling young α-MNs, whereas protocols of a higher tracer concentration and longer labeling duration were generally better in labeling aged α-MNs. In conclusion, the results of this systematic assessment provide a useful guide for the selection of optimal FB or CTB protocols for labeling young and aged α-MNs in normal, aging, and neurodegenerative disease studies.

## Figures and Tables

**Figure 1 bioengineering-10-00141-f001:**
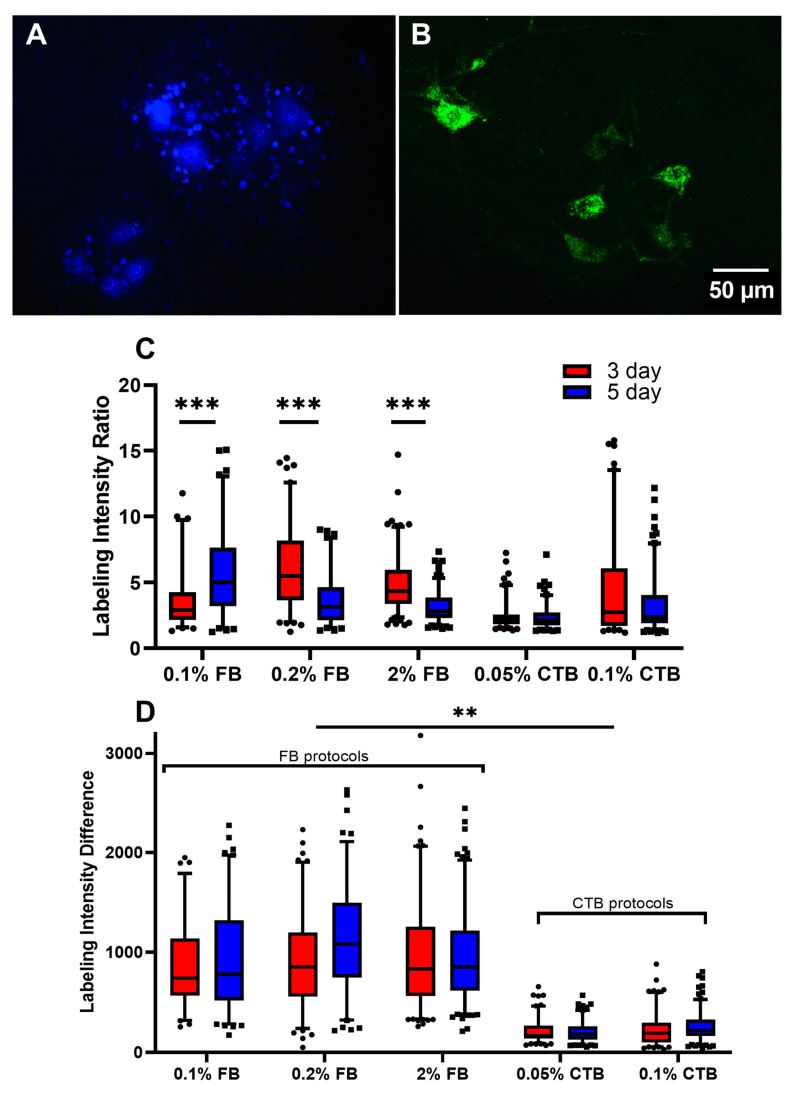
Neuronal labeling intensity ratio and difference among tracer protocols of young mice (6–7 weeks). (**A**) An FB (2%, 5-day labeling intensity protocol) image taken from the tissue of a young mouse. (**B**) A CTB (0.1%, 5-day labeling intensity protocol) image taken from the tissue of a young mouse. (**C**) Intensity ratios (labeling/background intensity). The number of cells analyzed per group (from left to right in order) is 75, 83, 102, 87, 145, 155, 30, 30, 30, and 30. (**D**) Intensity difference for all experimental groups measured 4 days after fixation. The number of cells analyzed per group (from left to right in order) is 72, 83, 102, 87, 145, 155, 30, 30, 30, and 30. Data are median ± 95% confidence interval. The circles and rectangles are data points outside the 95% confidence interval for the 3-day and 5-day protocols, respectively. *** denotes *p* < 0.001, ** denotes *p* < 0.01.

**Figure 2 bioengineering-10-00141-f002:**
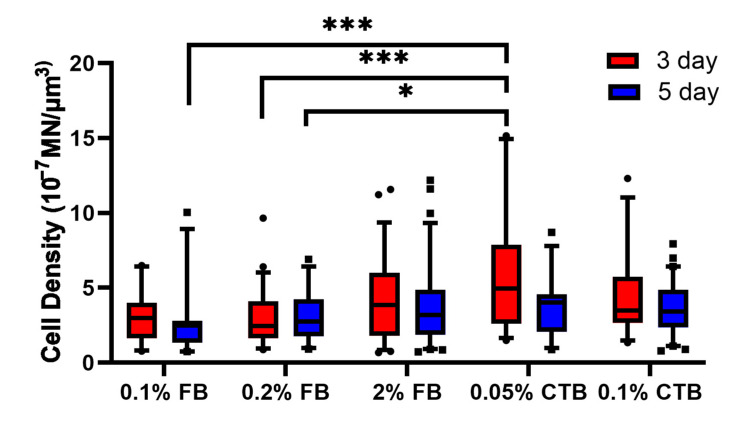
Density of labeled α-ΜΝs among tracer protocols. α-ΜΝ density was measured as the number of labeled MNs per unit tissue volume from 3D z-stack images for all experimental groups of young mice (6–7 weeks). The number of cells analyzed per group (from left to right in order) is 73, 97, 168, 97, 272, 277, 169, 139, 130, and 260. Data are median ± 95% confidence interval. The circles and rectangles are data points outside the 95% confidence interval for the 3-day and 5-day protocols, respectively. *** denotes *p* < 0.001 and * denotes *p* < 0.05.

**Figure 3 bioengineering-10-00141-f003:**
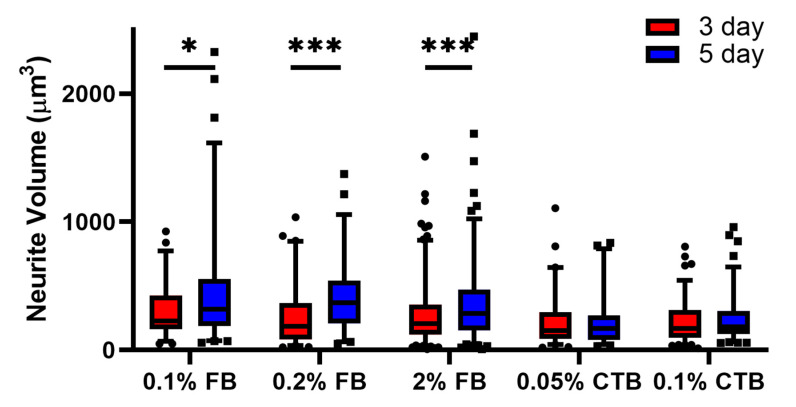
Neurite volume measurements among tracer protocols of young mice (6–7 weeks). The number of cells analyzed per group (from left to right in order) is 50, 67, 61, 48, 162, 126, 62, 53, 99, and 99. Data are median ± 95% confidence interval. The circles and rectangles are data points outside the 95% confidence interval for the 3-day and 5-day protocols, respectively. *** denotes *p* < 0.001 and * denotes *p* < 0.05.

**Figure 4 bioengineering-10-00141-f004:**
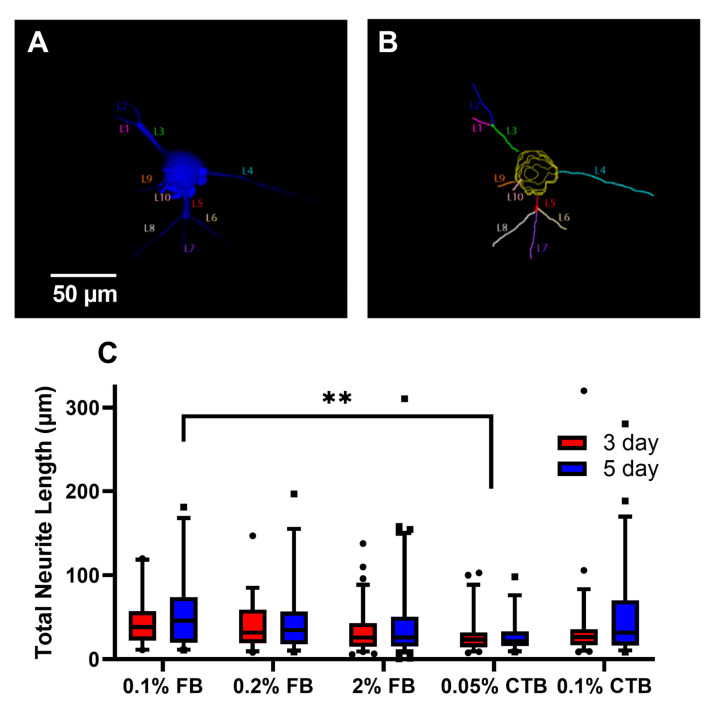
Total neurite length among tracer protocols of young mice (6–7 weeks). (**A**) Image of neurite projection prior to reconstruction from tracer labeling. Image from Neurolucida. (**B**) Image of neurite projections reconstructed from tracer labeling. The total neurite length was calculated as the sum of L1, L2, …, L10. (**C**) Total neurite length among tracer protocols. The number of cells analyzed per group (from left to right in order) in (**C**) is 27, 34, 39, 32, 75, 80, 45, 36, 55, and 48. Data are median ± 95% confidence interval. The circles and rectangles are data points outside the 95% confidence interval for the 3-day and 5-day protocols, respectively. ** denotes *p* < 0.01.

**Figure 5 bioengineering-10-00141-f005:**
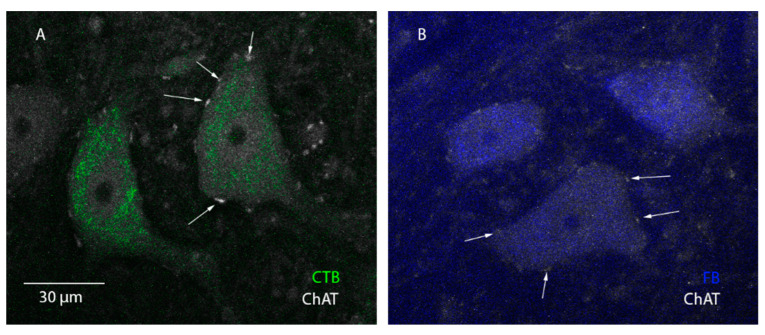
Co-labeling of ChAT with FB or CTB. 60× images of MNs labeled with 0.1% CTB 3-day (**A**) and 2% FB 3-day (**B**) co-labeled with ChAT. White arrows indicate the location of C-bouton labeling. The scale bar represents 30 µm. Image taken on Olympus FV1000 confocal.

**Figure 6 bioengineering-10-00141-f006:**
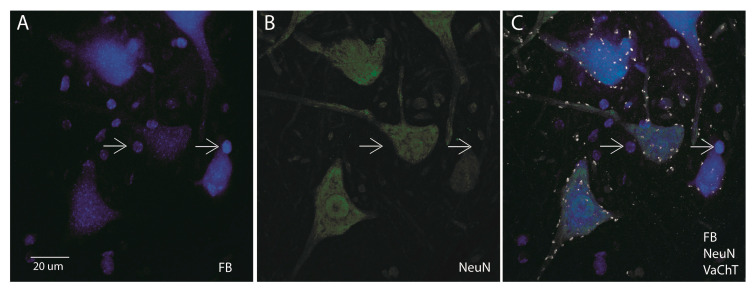
Tracer leakage in 2% FB 3-day protocol. A 60× image of (**A**) FB labeling, (**B**) NeuN labeling, and (**C**) FB, NeuN, and VAChT labeling. The white arrows indicate non-neuronal structures that are positive for FB+ (in (**A**)) but negative for both NeuN (in (**B**)) and VAChT (in (**C**)). The scale bar represents 20 µm. Image taken on Olympus FV1000 confocal.

**Figure 7 bioengineering-10-00141-f007:**
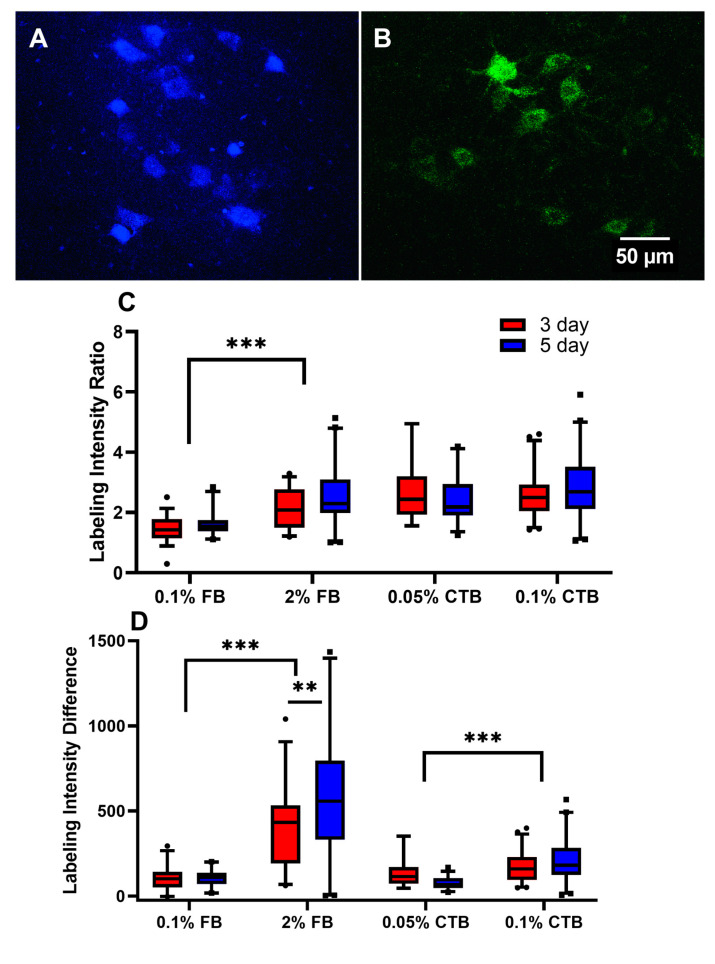
Neuronal label intensity ratio and difference among aged mice (25 months) tracer protocols. (**A**) An FB (2%, 5-day labeling duration protocol) image taken from the tissue of an old mouse. (**B**) A CTB (0.1%, 5-day labeling duration protocol) image taken from the tissue of an old mouse. (**C**) Intensity ratios (labeling/background intensity). The number of cells analyzed per group (from left to right in order) is 36, 24, 30, 42, 18, 28, 45, and 44. (**D**) Intensity difference for all experimental groups. The number of cells analyzed per group (from left to right in order) is 36, 24, 45, 44, 18, 28, 30, and 42. Data are median ± 95% confidence interval. The circles and rectangles are data points outside the 95% confidence interval for the 3-day and 5-day protocols, respectively. *** denotes *p* < 0.001, ** denotes *p* < 0.01.

**Figure 8 bioengineering-10-00141-f008:**
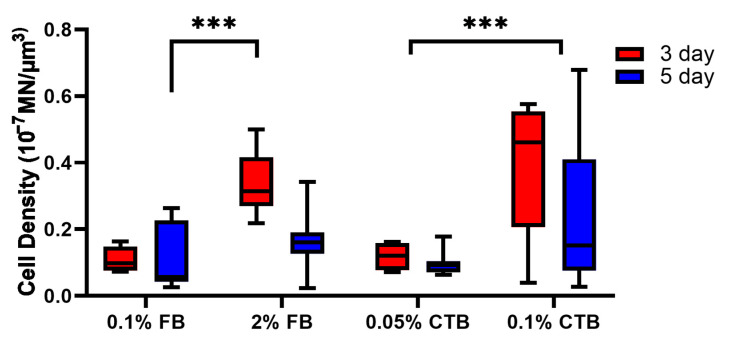
Density of labeled α-MNs in aged mice (25 months) among tracer protocols. α-ΜΝ density was measured as the number of labeled MNs per unit tissue volume from 3D z-stack images for all experimental groups. The number of cells analyzed per group (from left to right in order) is 8, 7, 8, 13, 4, 7, 8, and 14. Data are median ± 95% confidence interval. *** denotes *p* < 0.001.

**Figure 9 bioengineering-10-00141-f009:**
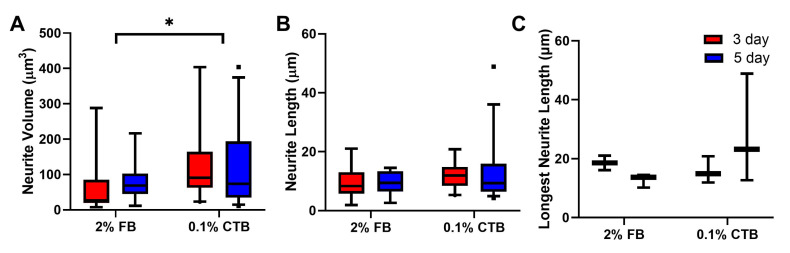
Total neurite volume, length, and longest path in aged mice (25 months). (**A**) Total neurite volume among tracer protocols. (**B**) Total neurite length of aged cells among the two tracer protocols. The total neurite length was calculated as the sum of L1, L2, …
, L10. (**C**) Longest neurite path of the two tracer protocols. Data are median ± 95% confidence interval. The rectangles are data points outside the 95% confidence interval for the 5-day protocols. * denotes *p* < 0.05.

**Table 1 bioengineering-10-00141-t001:** Young WT (B6SJL, 6–7 weeks) and aged (C57Bl/6, 25 months) male mice were assigned to different tracer protocols each representing a given tracer (FB or CTB) at a given concentration (% weight/volume) and labeling duration (3-day or 5-day).

Tracer	Concentration (%)	Labeling Duration	Sex	# of Young B6SJL Mice	# of Aged C57 Mice
CTB	0.05%	3-day	Male	3	3
		5-day	Male	3	3
	0.1% (control)	3-day	Male	3	3
		5-day	Male	4	3
FB	0.1%	3-day	Male	3	3
		5-day	Male	3	3
	0.2%	3-day	Male	3	0
		5-day	Male	3	0
	2% (control)	3-day	Male	3	3
		5-day	Male	3	3

**Table 2 bioengineering-10-00141-t002:** NeuN Co-labeling Analysis. FB sections from 0.1% FB 5-day, 0.2% FB 5-day, 2% FB 3-day, and 2% FB 5-day were stained with NeuN and VAChT to determine if blue, fluorescent dots were neuronal and/or motoneurons.

Protocol	# of Non-MN FB Dots	# of Co-Labeled FB Dots w/NeuN	% Co-Labeled
0.1% FB 5-day	28	5	17.85
0.2% FB 5-day	11	7	63.63
2% FB 3-day	56	4	7.14
2% FB 5-day	87	3	3.44

**Table 3 bioengineering-10-00141-t003:** Comparison of the tracer protocols. L and H refer to low and high concentrations and S and L refer to short and long labeling durations, respectively.

Age	Successful Protocol	Intensity		Cell Density	Neurite			Avoid Leakage
		Ratio	Difference		Volume	Length	Longest Path	
Young	Tracer	FB/CTB	FB	CTB	FB	FB	FB/CTB	CTB/FB
	Concentration	L/H	L/H	L/H	L/H	L/H	L/H	CTB (any) FB (L)
	Labeling duration	S/L	S/L	S/L	L	S/L	S/L	CTB (any) FB (S)
Aged	Tracer	FB/CTB	FB	FB/CTB	CTB	FB/CTB	FB/CTB	not assessed
	Concentration	H	H	H	H	H	H	not assessed
	Labeling duration	L	L	S/L	S/L	S/L	S/L	not assessed

## Data Availability

The data presented in this study are available upon reasonable request from the corresponding author.

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
