# Peer review of "Fast Blue and Cholera Toxin-B Survival Guide for Alpha-Motoneurons Labeling: Less Is Better in Young B6SJL Mice, but More Is Better in Aged C57Bl/J Mice"

_bioengineering, 2023, doi:10.3390/bioengineering10020141_

Round 1

Reviewer 1 Report

The manuscript entitled: "FB and CTB Survival Guide for Alpha-Motoneurons Labeling: Less is Better in Young Mice, but More is Better in Aged Mice" studies an important issue among neuroscientists who use tracers in order to identify numerous neurons in CNS. It is well written and provides evidence and comparison on many aspects of the labeled neurons.

However, in a study of this importance, I would like to see additional information that I am describing in the following paragraphs.

Regarding Animals 2.1.

Please advise the ARRIVE guidelines in the website provided below:

ARRIVE guidelines: https://arriveguidelines.org/sites/arrive/files/documents/ARRIVE%20guidelines%202.0%20-%20English.pdf

a. Provide species-appropriate details of the animals used, including species, strain and substrain, sex, age or developmental stage, and, if relevant, weight.
b. Provide further relevant information on the provenance of animals, health/immune status,
genetic modification status, genotype, and any previous procedures.

Also, it has to be said why the study used only male mice.

https://www.lasa.co.uk/inclusion-of-both-sexes-in-experimental-design-public-launch-from-the-mrc/

Table 1, shows that the aged mice were separated in 8 groups. Please explain why

Please explain which are the control animals in your study.

 For each experimental group, including controls, describe the procedures in enough detail to allow others to replicate them, including:
a. What was done, how it was done and what was used.
b. When and how often.
c. Where (including detail of any acclimatisation periods).
d. Why (provide rationale for procedures).

Please provide why aged animals received two surgeries and how this fact was accepted by the ethical committee. Lines 110-111.

Regarding Results

All Figures in Results e. g. in Fig. 3: the number of cells with different concentrations of the two tracers were different: 50, 67, 61, 48, 162, 126, 62, 53, 99 and 99. These are the lower numbers inside the red or blue boxes. Please explain. How we can compare the volume measurement of 61 versus 162 measured cells?

Mean value of the measurements are compared, but I would like to see more explanation on this issue.

Some general comments:

lines 82-87

 Our results showed that tracer protocols of lower concentrations and shorter survival days were generally better in labeling young α-MNs, whereas tracer protocols of higher tracer concentrations and longer survival days were generally better in labeling aged α-MNs. A low concentration, short survival FB protocol  provided optimal labeling of young α-MNs, whereas a high concentration, long survival  FB or CTB protocol provided optimal labeling of aged α-MNs.

Repetition in these two sentences

Line 150, lumbar instead of lumber

Line 151, stained instead of painted

Line 398, please provide explanation of LCA acronym

Line 622, please provide explanation of NJM acronym

General: VaCHT is used more. Some VaCht also appear in the text. Please select one of the two acronyms.

Author Response

Please see the attached file for our responses to reviewer #1

Reviewer 2 Report

I have checked the paper and it seems very well written and experiments comprehensively analyzed. However, I have serious doubts about its fitting with the scope of the journal. The paper describes a motor neuron labeling technology that seems more appropriate to another type of journal which is more focused on methodological issues, or alternatively to a journal of the Neuroscience field.   Throughout the manuscript there is no reference at all to stem cells, biomaterials, functionalizing factors, or cell differentiation, which are the pillars of Tissue Engineering and, this is why I believe it does not fit with the scope of Bioengineering journal.

Therefore, I'd suggest its transfer to another journal with a more appropriate scope: Methods in cell biology, Neuroanatomy, or Neuroscience.

Author Response

Please see the attached file for our responses to reviewer #2

Reviewer 3 Report

This research article worked on the potential improvement of existing staining protocols for retrogradely labelling motoneurons with two commonly used strainers: Fast Blue and Cholera Toxin-B. The technique of retrograde labelling of neurons in the spinal cord has been routine; however, a broad range of the quality and especially cell-specificity of labelling exists between experiments, even more across different studies. Therefore, the scientific idea of this research article is rational and can be of interest to many researchers. However, the way of executing this study, in particular, data analysis and presentation of the results, raises many concerns. I can only evaluate the scientific content of this paper once all technical issues are resolved.

-       The authors provided detailed descriptions of all tissue sample procedures but not a sufficient description of the imaging protocols and where precisely the imaging was performed in the tissue. One can assume that the whole vernal horn area was visualised for the lower motor neurons in the spinal cord slices, but it is essential to specify the exact regions (laminae/specific nuclei). What was the size of the imaged area, the resolution used, etc.?

-       Further, were morphological criteria (somata size/diameter) applied to the labelled cells, which were then taken into analysis? This would clarify whether mainly motoneurons were taken into analyses or various cell types.

-       The parameters the authors used for the imaging analysis are not “independent” (e.g. line 19), as most of them relate to morphological traits – they can be “different”.

-  “Survival days” (lines 21, 97, 133, and many others) imply severe experimental procedures requiring specific post-op animal care. Did the authors mean it for “staining/retrograde labelling duration”? Please clarify.

-       Full names should be given in the title.

-       The Introduction requires more careful work on retrograde labelling. It would benefit from a description of the principles of retrograde labelling and synaptic permeability to understand the limitations of this approach (paragraph 1). It has a very limited use/rationale in cultures (the 2nd  paragraph).

- The “leakage” sounds odd in the case of retrograde tracers, which label all structures connected to where they were injected (lines 66, 283, and others).

-       To conclude the protocol's applicability in neurodegenerative disease conditions, the authors must study it using an animal model of neurodegeneration. Otherwise, it is an overestimation and repeated speculations in the abstract, introduction, other sections.

-       What did the authors take as the background in the tissue slices (lines 214, 218)?

-       I do not understand why the authors introduced two parameters of “labeling intensity ratio” and “labeling intensity difference”. It appears somewhat amateur, as both parameters are causally related unless an inappropriate background was chosen for image quantifications. The ratio here has high values, assuming the background was near zero, which barely happens in any stained tissue. The authors have to show images to understand the labelling profile (areas with staining and without).

-    Similar concerns about the parameter of the density. Motoneurons are quite large cells, and counting the number of cells through 3D images, not single-section images (across different focal planes), would be more logical. It is feasible as the authors collected Z-stacks.  

-       It is impossible to assess the staining analysis with no single image provided to show typical tissue staining! Please demonstrate the example images to compare with those of others visually. How homogenous was the staining quality across the field of view after improving staining protocols?

-       The only place where the authors specify the age of mice is the methods section, not even the results – making it clear across the text, starting from the abstract, would benefit the readers’ understanding of what exact age the authors mean.

-       There were two different mouse strains used for “young” and “aged” cohorts – therefore, the results cannot be directly compared.

-  For the non-parametrically distributed data, the medium values should be presented. All data plots need to be changed.   

Author Response

Please see the attached file for our responses to reviewer #3

Round 2

Reviewer 1 Report

The manuscript entitled: "FB and CTB survival guide for alpha-motoneurons labeling: less is better in young mice, but more is better in aged mice" has been greatly improved  following the comments of the first review.

However,  more clarification is necessary on a couple of issues:

1. the authors state that for figure 3, previous data collected in their lab determined that 40 cells are usually sufficient to reach an N that allows for enough power to determine volume differences between groups. Could they state why this is not necessary for the results depicted in figure 1, where less than 40 cells were used?

2. The authors also state that there is variability in the number of data points among the groups (due to experimental factors that we do not have control over). Could they please explain these reasons, since it is not easily understood why this happens.

3. Some figures, or parts of figures, e.g. D in figure 1 and figure 7, C in figure 9 etc. are showing box plots that are difficult to distinguish due to the shrunk lines. I do not know if there is a way to modify this in order for the lines to be more visible.

4. In the legend of Figure 3, it is stated that the numbers inside the bars...., but there are not numbers in the bars anymore. This statement is left from the previous version, incorrectly.

Author Response

The manuscript entitled: "FB and CTB survival guide for alpha-motoneurons labeling: less is better in young mice, but more is better in aged mice" has been greatly improved following the comments of the first review.

Authors’ Response #1: We are glad that the reviewer found the revised manuscript greatly improved!

However, more clarification is necessary on a couple of issues:

  1. the authors state that for figure 3, previous data collected in their lab determined that 40 cells are usually sufficient to reach an N that allows for enough power to determine volume differences between groups. Could they state why this is not necessary for the results depicted in figure 1, where less than 40 cells were used?

Authors’ Response #2: Because biological variability is different among cell parameters, different cell parameters have different minimum sample sizes to get enough power. The 40 cells minimum sample size was for the volume property in figure 3, but figure 1 is for labeling intensity, which is a different property whose minimum sample size, in our experience, is 25 cells. While the minimum needed sample size differed among cell properties, we always collected measurements to exceed that minimum needed sample size. We have clarified this point in the revised manuscript.

  1. The authors also state that there is variability in the number of data points among the groups (due to experimental factors that we do not have control over). Could they please explain these reasons, since it is not easily understood why this happens.

Authors’ Response #3: One primary factor that contributed to this variability in sample sizes among the tracer protocols is the labeling intensity, which affects how well we can collect accurate measurements from an image. Because the quality of cell labeling is proportional to the labeling intensity (i.e., how bright the labeling is), tracer concentrations that had high labeling intensity (i.e., labeling is bright in the images) usually allowed us to collect large sample size of measurements (because we were able to measure the different parameters accurately), whereas tracer concentrations with low labeling intensity (i.e., labeling is dim in the images) did not allow us to collect large sample size of measurements (because we had difficulty measuring the different parameters accurately). For instance, in young mice, the 3-day and 5-day FB 2% protocols labeled MNs with high intensity (see fig. 1C and D), whereas the 3-day and 5-day CTB 0.05% protocols labeled MNs with low intensity (see fig. 1C and D). Because of that difference in labeling quality, measurements from the FB 2% protocols usually had large sample size, whereas measurements from the CTB 0.05% protocols usually had small sample size. We clarified this point in the revised manuscript.

  1. Some figures, or parts of figures, e.g. D in figure 1 and figure 7, C in figure 9 etc. are showing box plots that are difficult to distinguish due to the shrunk lines. I do not know if there is a way to modify this in order for the lines to be more visible.

Authors’ Response #4: To address the reviewer’s comment, we tried to place a break in the y-axis of these figures to improve the visualization of the data. However, because some box plots had high values while the others had low values (figure 1D is an example), it was impossible to find a way to improve the visualization of some box plots without deteriorating the visualization of the others. Thus, we ended up adjusting the scale of the y-axes of many figures (fig. 1D, fig. 3, fig. 7D) to improve the visualization of these figures as much as possible.

  1. In the legend of Figure 3, it is stated that the numbers inside the bars...., but there are not numbers in the bars anymore. This statement is left from the previous version, incorrectly.

Authors’ Response #5: We apologize for overlooking this. We have removed this old text from the revised manuscript.

Reviewer 2 Report

Thank you for the clarification. For what I can tell as just an external reviewer, I believe that the journal’s editorial office invitation meant just that: you were invited to submit the paper for evaluation, but in my view that did not necessarily imply any assessment of the scientific content and scope of the manuscript. This is just my personal interpretation. For the rest, I remain convinced that this work does not fit well with the scope of this journal in particular. The manuscript is about a motor neuron labeling method which involves no bioengineering or tissue engineering procedures whatsoever. I must again recommend its transfer to another more appropriate specialized journal in Neuroanatomy or Neuroscience.

Author Response

Thank you for the clarification. For what I can tell as just an external reviewer, I believe that the journal’s editorial office invitation meant just that: you were invited to submit the paper for evaluation, but in my view that did not necessarily imply any assessment of the scientific content and scope of the manuscript. This is just my personal interpretation. For the rest, I remain convinced that this work does not fit well with the scope of this journal in particular. The manuscript is about a motor neuron labeling method which involves no bioengineering or tissue engineering procedures whatsoever. I must again recommend its transfer to another more appropriate specialized journal in Neuroanatomy or Neuroscience.

Authors’ Response #6: We would like to clarify to the reviewer that it was not an invitation that we received from the journal, but it was a request from us to the journal to know if this study fits the journal’s scope. The editorial office’s answer was yes. Regardless, we completely respect the reviewer’s opinion. We just wanted to clarify the situation and show that we attempted to get this question answered upfront before we submit the manuscript.

Reviewer 3 Report

The authors addressed some of the critical issues of the manuscript. However, both title and abstract assumed a generalized protocol use in 'young and aged animals', which later appears relevant to specific mouse strains. It must be clear from the very beginning.  

Scale bars are required for all the images shown, with a readable font size. 

Author Response

The authors addressed some of the critical issues of the manuscript. However, both title and abstract assumed a generalized protocol use in 'young and aged animals', which later appears relevant to specific mouse strains. It must be clear from the very beginning.  

Authors’ Response #7: We now mention the strains of young and aged mice in the title and the abstract to avoid the generalization.

Scale bars are required for all the images shown, with a readable font size. 

Authors’ Response #8: We have included readable scale bars in all the images in the revised manuscript.